# Sensitivity and robustness of sky-polarimetric Viking navigation: Sailing success is most sensitive to night sailing, navigation periodicity and sailing date, but robust against weather conditions

Péter Takács[1], Dénes Száz[1,2], Ádám Pereszlényi[1,3], Gábor Horváth[1]*

**1** Department of Biological Physics, Environmental Optics Laboratory, ELTE Eötvös Loránd University, Budapest, Hungary, **2** ELTE BDPK Department of Physics, Szombathely, Hungary, **3** Deutsches Meeresmuseum, Stralsund, Germany

* gh@arago.elte.hu

**Data Availability Statement:** All relevant data are within the manuscript.

## Abstract

Although Viking sailors did not have a magnetic compass, they could successfully navigate with a sun-compass under a sunny sky. Under cloudy/foggy conditions, they might have applied the sky-polarimetric Viking navigation (SPVN), the high success of which has been demonstrated with computer simulations using the following input data: sky polarization patterns measured with full-sky imaging polarimetry, and error functions of the navigation steps measured in psychophysical laboratory and planetarium experiments. As a continuation of the earlier studies, in this work we investigate the sensitivity of the success of SPVN to the following relevant sailing, meteorological and navigational parameters: sunstone type, sailing date, navigation periodicity, night sailing, dominance of strongly, medium or weakly cloudy skies, and changeability of cloudiness. Randomly varying these parameters in the simulation of Viking voyages along the latitude 60˚ 21' 55'' N from Norway to Greenland, we determined those parameters which had strong and weak influences on the success of SPVN. The following intrinsic parameters of the simulation were also randomly changed: sailing speed, visibility distance of Greenland's southeast coastline and start time of diurnal sailing. Our results show that the sailing success is sensitive to the night sailing, navigation periodicity and sailing date, while it is robust against the sunstone type, dominance of strongly, medium or weakly cloudy skies, and changing cloudiness.

## Introduction

Vikings navigated successfully on the North Atlantic Ocean for three hundred years without a magnetic compass [1–5]. Under sunny weather conditions, they might have navigated by a sun-compass and when the sun was occluded by clouds/fog or was below but near the horizon, they used this compass and the sky polarization detected by sunstone crystals [6–8]. Although

**Funding:** The author(s) received no specific funding for this work.

**Competing interests:** The authors have declared that no competing interests exist.

all aspects of this hypothetical sky-polarimetric Viking navigation (SPVN) were purely speculative, many scientists [9–13] accepted and cited it as if it were a fact. Some sceptic researchers, however, expressed contra-arguments [14] disputing why this navigation method could not have functioned under certain meteorological conditions. Between the two groups of believers and sceptics, a third group was formed by researchers who tried to reveal which components/ aspects of this hypothesis may be valid or unfounded. Ropars *et al.* [15] studied a calcite crystal that might have been a precise depolarizing sunstone. Le Floch *et al.* [16] analysed the sixteenth century Alderney calcite crystal that could be an efficient reference optical Viking compass.

In the last decades the most thorough and systematic studies of SPVN have been performed by a Hungarian group cooperating with German, Swedish and Swiss researchers: They measured the polarization patterns of the sky with full-sky imaging polarimetry, and investigated the atmospheric optical and meteorological prerequisites of SPVN [17–26]. The error functions of the steps of SPVN have been measured on numerous test persons in psychophysical laboratory and planetarium experiments [27–31]. Száz *et al.* [31] determined the accuracy of SPVN as functions of the solar elevation and cloudiness. Bernáth *et al.* [32] proposed an alternative interpretation of the Uunartoq (Viking) sundial artefact and suggested that it might have been an instrument to determine the latitude and local noon. Bernáth *et al.* [33] also interpreted the Uunartoq fragment as a twilight board and demonstrated in a field experiment how the Viking sun-compass could have been used with sunstones before and after sunset. Using celestial polarization patterns and psychophysically measured errors, Száz and Horváth [34] revealed with computer simulations the chance that Vikings could reach Greenland.

After these studies the logical question of the sensitivity or robustness of SPVN has arisen. Using an improved version of the software of Száz and Horváth [34], in this work we investigate the sensitivity/robustness of the success of SPVN to the following relevant sailing, meteorological and navigational parameters: sunstone crystal type, sailing date, navigation periodicity, sailing or staying at night, dominance of strongly, medium or weakly cloudy skies, and cloudiness changeability. Randomly varying these parameters in the simulations of Viking voyages, we determined those parameters which have strong and weak influences on the success of SPVN. The sailing speed and start time of diurnal sailing were also randomly changed, and the visibility distance of Greenland's southeast coastline depended on the current cloudiness situation.

In this work, the terms 'navigation' and 'sailing' are used in the following contexts: *Navigation* means the four-step process of SPVN, during which a Viking navigator determines the geographical north and then the ship's intended moving direction. *Sailing* means the journey of a Viking ship, the navigator of which performs numerous navigation processes.

## Methods

### Computational methods

We simulated 1 000 000 Viking voyages from Bergen (Norway) to Hvarf (Greenland). The route of the Viking voyage between Norway and south Greenland (Hvarf) ran along the 60˚ 21' 55" northern latitude and started from Hernam (nowadays the Norwegian Bergen) [2, 3]. One of the possible explanations of this latitude could be that the prevailing ocean currents might have been taken into consideration by Vikings. On the other hand, this straight route was the shortest between Norway and Hvarf. If the departure had not been from Bergen, but much further north or south in Norway, then the duration of a voyage would have been longer, and other shapes of the gnomonic lines should have been used as the straight (for spring equinoxes) or hyperbolic (for summer solstices) lines found on the sun-compass artefacts. Our

 

computations had the following five internal parameters: solar elevation, north error, sailing speed, start time of diurnal sailing, and visibility distance of Greenland's southern coastline. The sensitivity/robustness of the successful SPVN was determined for the following six variables: sailing date, sunstone type, night sailing, cloudiness dominance, cloudiness changeability, and navigation periodicity. Before each run of a simulated voyage, the values of the following parameters were chosen randomly and independently of each other:

- sailing date: spring equinox or summer solstice

- sunstone type: calcite, cordierite or tourmaline

- night sailing: yes or no

The spring equinox (21 March) and the summer solstice (21 June) meant two important dates of the Viking sailing season: the former and latter was approximately the start and the middle of this season. In our simulations we chose only the spring equinox and the summer solstice as sailing dates, because the two artefacts of the Viking sun-compass (discovered in the Vatnahverfi and Uunartoq Fjords of Greenland) had only two clearly discernible gnomonic lines (trajectories of the tip of the shadow cast by the vertical gnomon on the horizontal disc of the sun-compass in sunshine) [3]: a straight line for navigation at spring equinoxes and a hyperbolic line for summer solstices. For other sailing dates the gnomonic lines are different hyperbolic curves.

The values of the following continuous parameters were also chosen randomly from a uniform distribution:

- cloudiness dominance: $-1 \leq m_{\mathrm{dominance}} \leq +1$

- cloudiness changeability: $0 \leq \sigma_{\mathrm{changeability}} \leq 4$

- navigation periodicity: 0.5 hour $\leq \Delta t \leq$ 6 hours

The temporal step of each simulated voyage was $\varepsilon$ = 1 minute, while the cloudiness $\rho$ was changed every 60 minutes. During an $\varepsilon$ period, the ship advanced $\Delta \underline{s} = \underline{w}\varepsilon$, where $\underline{w}$ is the velocity vector of the ship in the current step. We performed the simulations with a custom-developed Python script. The variables during the simulations were the following:

**Sailing date: Spring equinox or summer solstice.** Voyages were simulated at spring equinox (21 March) and summer solstice (21 June). Before each run we randomly selected the sailing date.

**Sunstone type.** In the simulations, we used calcite, cordierite and tourmaline sunstones, for which the error functions of the steps of SPVN have been published earlier [28, 30]. Before each run, we randomly selected a crystal, which was used during the full voyage.

**Solar elevation.** Always the actual solar elevation $\theta$ was used in every point of time of the sailing. Solar elevation data were taken from the following web site: https://www.sunearthtools.com/dp/tools/pos_sun.php.

**Navigation periodicity.** During a given time period $\tau$, the simulated Viking ship moved in a constant direction along a straight line with a constant sailing speed $w$ until the navigator determined the new sailing direction. In the simulation, the time segment $\tau$ was chosen randomly with a uniform distribution from the interval $\Delta t - \Delta t/6 \leq \tau \leq \Delta t + \Delta t/6$, where $\Delta t$ is the navigation periodicity (= the elapsed time since the last navigation) [34]. Before each run, $\Delta t$ was randomly selected from the following range: 0.5 hour $\leq \Delta t \leq$ 6 hours.

**Dominance and changeability of cloudiness.** Száz and Horváth [34] assumed an equal distribution/chance of good (with cloudiness $0 \leq \rho < 3$ oktas) and bad ($5$ oktas $< \rho \leq 8$ oktas) weathers during their simulated Viking voyages. This assumption is considered here as a

 

medium (normal) weather with cloudiness dominance $m_{dominance}$ = 0 meaning a medium cloudy sky. We tried to simulate the temporal change of cloudiness as realistic as possible. The cloudiness ρ ranged from 0 to 8 oktas (0 okta: no clouds, 8 oktas: completely overcast sky), where okta means 1/8 area of the sky hemisphere. The first ρ-value at the start of a simulated voyage was chosen from a Gaussian distribution with a median of 4 and a deviation of 2. Each succeeding ρ-value was calculated by adding a discrete change (increment) in okta to the previous ρ-value according to a second Gaussian distribution with median $m_{dominance}$ and deviation $σ_{changeability}$, where $m_{dominance}$ denotes the dominance of clouds on the sky (-1 ≤ $m_{dominance}$ ≤ -0.75 for weakly cloudy skies, -0.125 ≤ $m_{dominance}$ ≤ +0.125 for medium cloudy skies, +0.75 ≤ $m_{dominance}$ ≤ +1 for strongly cloudy skies) and $σ_{changeability}$ denotes the cloudiness changeability (the higher the $σ_{changeability}$, the higher the changeability of oktas in the next hour), so that ρ had to remain in the range of 0 and 8 oktas: for ρ < 0 okta or ρ > 8 oktas, the cloudiness was set to 0 and 8 oktas, respectively. The cloudiness ρ was recalculated hourly.

**North error.** The angle of deviation from the geographical north, called north error, was determined as follows: For a given solar elevation angle θ we calculated the actual cloudiness ρ as described above, then we used the 1080 data files from the θ-ρ matrix determined in [31]. In this matrix, every θ-ρ pair contains 12 different sky situations with a known north error distribution calculated with the use of the errors of the four steps of SPVN measured in psychophysical experiments. One of these 12 sky situations was selected randomly according to uniform distribution, then from the corresponding error distribution we used a randomly chosen error value as the actual north error.

**Sailing speed.** We characterized the ship's sailing speed $w$ (which was constant 11 km/h in [34]) with three parameters: (i) maximum speed $w_{max}$ = 21 km/h, (ii) average speed $w_{ave}$ = 11 km/h, and (iii) speed's standard deviation $Δw_{ave}$ = 2 km/h. For each change of the north error the simulation generated a new speed value.

**Night sailing.** Considering the night sailing, in the simulations of sailing routes there were two options:

1. Between sunset and sunrise, Vikings lowered their sails and stopped their voyage as in [34]. In this case, at night the ship's position was constant. With a non-lowered sail, the ship would have randomly drifted due to the random winds, which drift would have made difficult the safe daytime continuation of the voyage. With a lowered sail, the ship did not drift at night. The influence of water currents and wind was neglected, because the average water speed in the Atlantic Ocean is 0.54–0.72 km/h [35]. On the other hand, the average wind speed is more than ten times larger (50–60 km/h, [35]).

2. At night, the ship continued its route in the last direction determined by the navigator at sunset.

Before the start of a simulated voyage, we randomly turned on or off the night sailing option.

**Start time of diurnal sailing.** The start time of diurnal sailing was selected randomly on every sailing day after sunrise within the range 0 and 1 hour from a uniform distribution. Note that in [34] the navigator started to navigate always at sunrise with zero sun elevation.

**Visibility distance of Greenland's southeast coastline (end of simulation).** A given simulation was stopped, when the ship reached the northeast coastline of North-America (unsuccessful sailing route) or the visibility distance $d$ of the southeast coastline of Greenland's south tip (successful sailing route), depending on the actual cloudiness value ρ (measured in okta) as follows: $d(ρ)$ = -16.02875·ρ + 128.23 km. This formula is derived in [34]. Assuming that in overcast and foggy weather with ρ = 8 the visibility distance $d(ρ = 8)$ = 0 km, the $d(ρ)$ function

was calculated with a linear interpolation between $d(\rho = 8) = 0$ km and $d(\rho = 0) = 128.23$ km. The linear function $d(\rho)$ modelled how the cloudiness may affect the visibility distance of the coastline, where $\rho$ changed stochastically. The map of the North Atlantic region was generated by a software written by us. The contours of continents and islands were manually digitalized from the map available as open-source data from http://www.gnuplotting.org/plotting-the-world-revisited/ (the raw data points of the contours can be freely downloaded in text format from: http://www.gnuplotting.org/data/world_10m.txt). These open-source data can be freely used without permission/licence. The coastline of Greenland (Fig 1) as the direct goal of voyages became limited to a southeast section, instead of the full east borderline in [34]. When a ship reached shore on the northern coastline of Greenland where there was no Viking settlement, the voyage was considered unsuccessful.

## Statistical analyses

We used logistic regression to obtain the significant variables determining the probability $\pi$ of successful SPVN as functions of the variables sailing date, sunstone type, cloudiness dominance $m_{dominance}$, cloudiness changeability $\sigma_{changeability}$, navigation periodicity $\Delta t$ and night sailing. Száz and Horváth [34] showed that the sailing success does not change monotonously with increasing $\Delta t$. Therefore, for the logistic regression we divided the continuous variable $\Delta t$ into 100 equal intervals. In the 1st logistic model we supposed interferences between $m_{dominance}$ and $\sigma_{changeability}$, as well as between sunstone type, $\Delta t$ and night sailing. We split the simulation results to train (900 000 simulations) and test (100 000 simulations) datasets. Using the train dataset, we built the 1st logistic model and determined the significant variables by applying ANOVA test for this model. The test dataset was used to calculate the accuracy of the model. On the basis of the results of the 1st model, we applied a 2nd logistic regression (model) with only the significant variables to determine the probability of successful sailing under different conditions. Statistical analyses were performed with the R statistics package 3.6.3. [36].

## Results

### Some visualized Viking voyages

Fig 1 visualizes some simulated Viking sailing routes from Bergen (Norway) to Hvarf (Greenland) at summer solstice using cordierite with navigation periodicity $\Delta t = 1$, 3 and 6 hours without and with night sailing. The sailing success rate $s$ is larger for $\Delta t = 1$ hour (without night sailing: $s = 99.8\%$, with night sailing: $s = 99.8\%$) than for $\Delta t = 6$ hours (without night sailing: $s = 0.0\%$, with night sailing: $s = 24.3\%$), independently of the night sailing. However, for $\Delta t = 3$ and 6 hours, with night sailing $s$ ($s_{\Delta t = 3h} = 100.0\%$, $s_{\Delta t = 6h} = 24.3\%$) is larger than without night sailing ($s_{\Delta t = 3h} = 19.0\%$, $s_{\Delta t = 6h} = 0.0\%$). Fig 1 demonstrates that night sailing has a significant influence on the sailing success, that is explained in the Discussion.

### Robust parameters having minimal effect on the sailing success

**Sunstone type.** Fig 2A shows the success rate of simulated voyages at summer solstice (21 June) as a function of the navigation periodicity $\Delta t$ for calcite, cordierite and tourmaline sunstones without night sailing. The sunstone type has only a minor effect on the sailing success rate $s$. The effect of sunstone type on $s$ is similarly negligible for different sailing dates, independently of night sailing. Száz and Horváth [34] obtained that the accuracy of sunstone adjustment in the first step of SPVN slightly depends on the crystal type. Here, we demonstrate

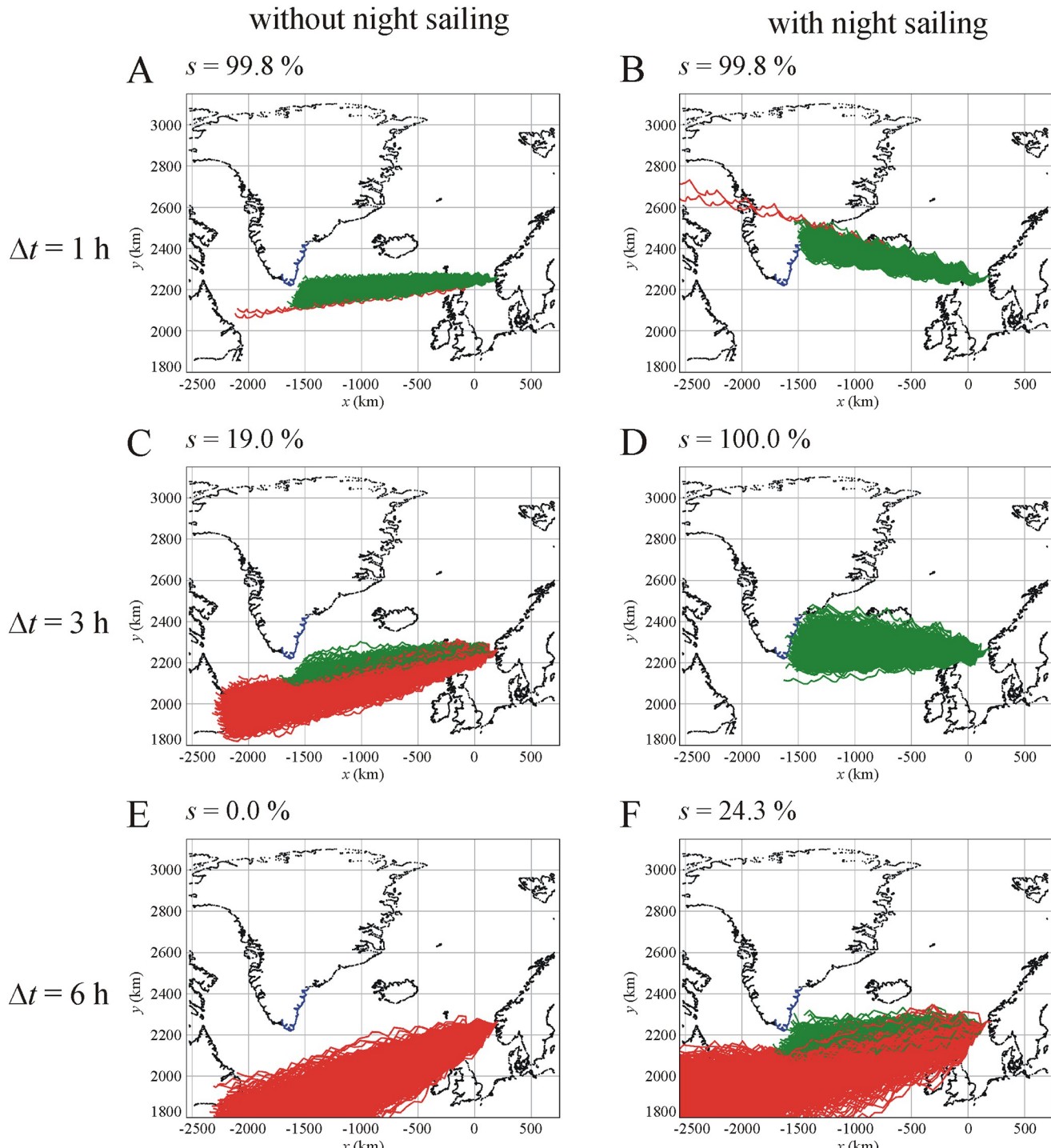

**Fig 1. Examples for simulated sailing routes using SPVN from Norway to Greenland.** Successful (green) and unsuccessful (red) routes of 1000 Viking voyages from Bergen (Norway) to Hvarf (Greenland) at summer solstice using cordierite sunstone with navigation periodicity $\Delta t = 1$ hour (A, B), $\Delta t = 3$ hours (C, D) and $\Delta t = 6$ hours (E, F) without (A, C, E) and with (B, D, F) night sailing. The values of the sailing success rate $s$ are: (A) 99.8%, (B) 99.8%, (C) 19.0%, (D) 100.0%, (E) 0.0%, (F) 24.3%. The blue curve is the borderline of visibility from which the southeast mountains of Greenland can already be seen from a Viking ship. The simulation of a voyage stops when the navigator sees the southeast coastline where the visibility distance is determined by the current cloudiness value ρ. Some simulated sailing trajectories pass through Iceland and/or North Scotland. In these cases, it was assumed that the Vikings continued their voyage towards Greenland. The maps are generated by our software after a manual selection of the contours of continents and islands from the open-source data originating from http://www.gnuplotting.org/plotting-the-world-revisited/.

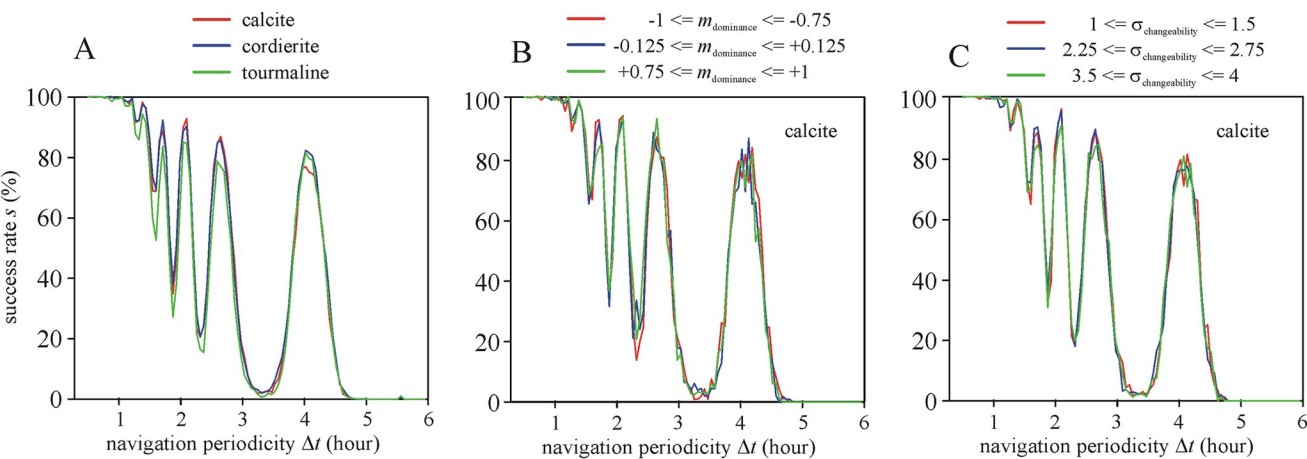

**Fig 2. Success rate *s* of simulated voyages using SPVN at summer solstice (21 June) as a function of the navigation periodicity Δ*t* without night sailing.** (A) *s*(Δ*t*) for calcite, cordierite and tourmaline sunstones. (B) *s*(Δ*t*) with calcite sunstone for weakly cloudy (-1 ≤ $m_{\mathrm{dominance}}$ ≤ -0.75), medium cloudy (-0.125 ≤ $m_{\mathrm{dominance}}$ ≤ +0.125) and strongly cloudy (+0.75 ≤ $m_{\mathrm{dominance}}$ ≤ +1) skies. (C) *s*(Δ*t*) with calcite sunstone for weak/slow (1 ≤ $\sigma_{\mathrm{changeability}}$ ≤ 1.5), medium (2.25 ≤ $\sigma_{\mathrm{changeability}}$ ≤ 2.75) and strong/rapid (3.5 ≤ $\sigma_{\mathrm{changeability}}$ ≤ 4) cloudiness changeabilities of the cloudiness ρ between simulation steps. The sunstone type, cloudiness dominance $m_{\mathrm{dominance}}$ and cloudiness changeability $\sigma_{\mathrm{changeability}}$ have only a minor influence on *s*, thus SPVN is robust against these parameters.

that the sunstone type has a negligible influence on the success rate *s* of voyages using four-step SPVN. Thus, SPVN is robust against the sunstone type.

**Cloudiness dominance.** Fig 2B displays the success rate *s* of voyages using calcite sunstone at summer solstice (21 June) without night sailing as a function of the navigation periodicity Δ*t* for weakly cloudy (-1 ≤ $m_{\mathrm{dominance}}$ ≤ -0.75), medium cloudy (-0.125 ≤ $m_{\mathrm{dominance}}$ ≤ +0.125) and strongly cloudy (+0.75 ≤ $m_{\mathrm{dominance}}$ ≤ +1) skies. The effect of $m_{\mathrm{dominance}}$ variable is negligible on *s*. The influence of cloudiness dominance on *s* is similarly minimal for different sailing dates, independently of night sailing. Hence, SPVN is robust against the cloudiness dominance $m_{\mathrm{dominance}}$.

**Cloudiness changeability.** Fig 2C illustrates the success rate *s* of voyages using calcite sunstone at summer solstice (21 June) without night sailing as a function of the navigation periodicity Δ*t* for weak/slow (1 ≤ $\sigma_{\mathrm{changeability}}$ ≤ 1.5), medium (2.25 ≤ $\sigma_{\mathrm{changeability}}$ ≤ 2.75) and strong/rapid (3.5 ≤ $\sigma_{\mathrm{changeability}}$ ≤ 4) change possibilities $\sigma_{\mathrm{changeability}}$ of the cloudiness ρ between simulation steps. $\sigma_{\mathrm{changeability}}$ has only a minor effect on *s*. The effect of sunstone on *s* is similarly negligible for different sailing dates, independently of night sailing. Consequently, SPVN is robust against the cloudiness changeability $\sigma_{\mathrm{changeability}}$.

## Sensitive parameters significantly influencing the sailing success

**Sailing date.** Fig 3 illustrates the success rate *s* of simulated voyages using SPVN with calcite sunstone at spring equinox (21 March) and summer solstice (21 June) as a function of the navigation periodicity Δ*t* with and without night sailing. Our simulations show a strong dependence of *s* on the sailing date: The sailing successes at spring equinox and summer solstice differ significantly from each other, independently of the night sailing. At summer solstice, the length of the day suitable for SPVN is longer than at equinox, thus the advancement of the ship is definitely longer for the former date. If, however, we assume continuous advancement of the ship at night based on the last set direction during the last navigation before

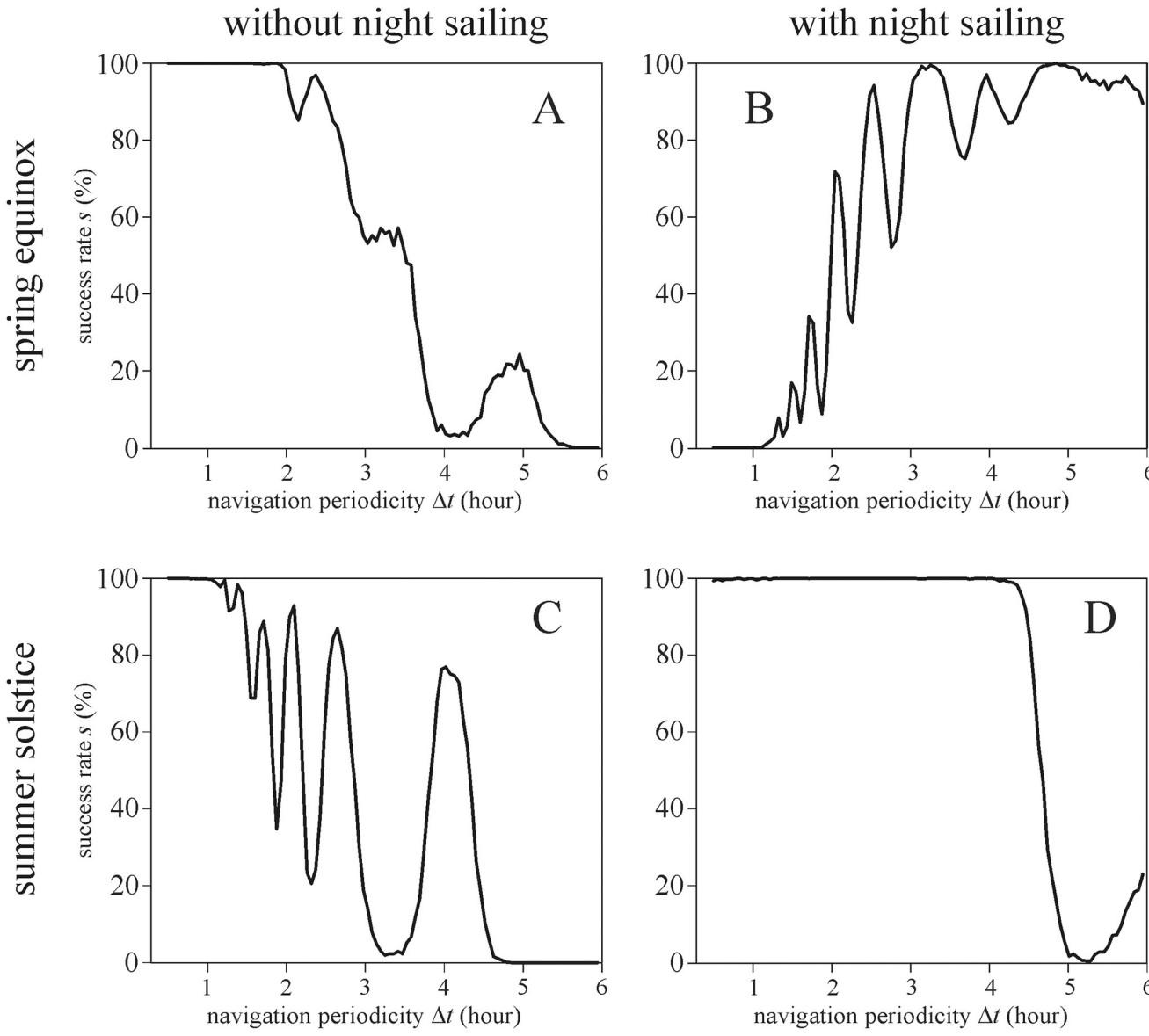

**Fig 3. Success rate s of simulated voyages using SPVN with calcite sunstones at spring equinox (21 March, A-B) and summer solstice (21 June, C-D) as a function of the navigation periodicity Δt without (A, C) and with (B, D) night sailing.** Since the effect of sunstone type, cloudiness dominance $m_{dominance}$ and cloudiness changeability $\sigma_{changeability}$ was negligible on s, we did not sort the data for these variables when created this figure.

sunset, the advancement time is equal for both dates. The significant differences between the sailing successes at spring equinox and summer solstice mean that the success s of SPVN is sensitive to the sailing date.

**Night sailing.** In their simulations, Száz and Horváth [34] assumed that the Viking ships did not sail during the night. In our present simulations, we tested the effect of night sailing on the sailing success. The success rate s strongly depended on night sailing. As Száz et al. [31] showed, if the navigation is in the afternoon, then the determined direction always points toward north. When night sailing was allowed, the Vikings sailed toward north all night (because even with Δt = 6 hours, the last navigation of the day always happened in the afternoon). If other parameters were set so that the ship deviated toward south (e.g. in summer

**Table 1. Results of the ANOVA test for the 1st logistic model.**

| variables | df | deviance | resid. df | resid. dev | p | significance |
|---|---|---|---|---|---|---|
| NULL | | | 999999 | 1348182 | | |
| crystal type | 2 | 264 | 999997 | 1347918 | < 2.2e-16 | *** |
| $m_{dominance}$ | 1 | 74 | 999996 | 1347844 | < 2.2e-16 | *** |
| $\sigma_{changeability}$ | 1 | 0 | 999995 | 1347843 | 0.496711 | |
| $\Delta t$ | 99 | 141068 | 999896 | 1206775 | < 2.2e-16 | *** |
| night sailing | 1 | 45475 | 999895 | 1161300 | < 2.2e-16 | *** |
| sailing date | 1 | 161 | 999894 | 1161139 | < 2.2e-16 | *** |
| $m_{dominance}$ & $\sigma_{changeability}$ | 1 | 8 | 999893 | 1161131 | 0.004813 | ** |
| $\Delta t$ & night sailing | 99 | 248383 | 999794 | 912748 | < 2.2e-16 | *** |
| $\Delta t$ & sailing date | 99 | 280489 | 999695 | 632260 | < 2.2e-16 | *** |
| night sailing & sailing date | 1 | 100488 | 999694 | 531772 | < 2.2e-16 | *** |
| $\Delta t$ & night sailing & sailing date | 99 | 18739 | 999595 | 513033 | < 2.2e-16 | *** |

df: degree of freedom, resid. df.: residual degree of freedom. The interacting variables are connected by &.

solstice with $\Delta t$ = 3 and 6 hours), night sailing could help to increase the success rate $s$ (see Fig 1C versus 1D and 1E versus 1F). However, for deviating too much toward north (i.e. anything above 65.37˚ latitude at Greenland's coastline), the sailing was regarded as unsuccessful. According to Fig 3, the sailing success rate $s$ versus $\Delta t$ is more or less chaotic, depending strongly on the initial conditions of the simulation (sailing date and night sailing). Thus, SPVN is sensitive to the night sailing.

**Navigation periodicity.** Fig 3 demonstrates that the sailing success rate $s$ strongly depends on the navigation periodicity $\Delta t$. Thus, SPVN is sensitive to $\Delta t$.

## Statistics

The accuracy of the 1st logistic model is 0.8874. According to Table 1, in this model the variables sailing date, crystal type, cloudiness dominance $m_{dominance}$, navigation periodicity $\Delta t$ and night sailing are highly significant ($p < 0.05$), while the cloudiness changeability $\sigma_{changeability}$ is not significant. The interacting variable pairs are also significant. The high significance of most of the variables is not surprising because of the high number of simulation (900 000 runs in the train dataset).

For the 2nd logistic model we used only the significant variables determined by the 1st logistic model: the sailing date, crystal type, cloudiness dominance $m_{dominance}$, navigation periodicity $\Delta t$ and night sailing. We supposed interactions between the following variables: sailing date, crystal type and navigation periodicity $\Delta t$. Table 2 contains the results of the ANOVA test applied for the 2nd logistic model. All variables and variable interactions show highly significant effects. The accuracy of the 2nd model is 0.8873, practically the same as that of the 1st model.

Figs 4–6 show the predicted probability $\pi$ of successful SPVN as functions of the significant variables. It is clearly seen that the crystal type and cloudiness dominance $m_{dominance}$ do not affect $\pi$. The shorter the navigation periodicity, the higher the $\pi$, but this effect is not monotonous as seen in Figs 4–6. However, if the voyage happens at spring equinox with night sailing, the longest navigation periodicity $\Delta t$ = 6 hours leads to higher $\pi$.

## Discussion

This work is a logical sequel of our previous investigation [34] in the topic of sky-polarimetric Viking navigation (SPVN). We provide a further elaboration of the potential success rate of

**Table 2. Results of the ANOVA test for the 2nd logistic model.**

| variables | df | deviance | resid. df | resid. dev | p | significance |
|---|---|---|---|---|---|---|
| NULL | | | 999999 | 1348182 | | |
| crystal type | 2 | 264 | 999997 | 1347918 | $< 2.2e\text{-}16$ | *** |
| $m_{\text{dominance}}$ | 1 | 74 | 999996 | 1347844 | $< 2.2e\text{-}16$ | *** |
| $\Delta t$ | 99 | 141067 | 999897 | 1206777 | $< 2.2e\text{-}16$ | *** |
| night sailing | 1 | 45475 | 999896 | 1161302 | $< 2.2e\text{-}16$ | *** |
| sailing date | 1 | 161 | 999895 | 1161141 | $< 2.2e\text{-}16$ | *** |
| $\Delta t$ & night sailing | 99 | 248378 | 999796 | 912763 | $< 2.2e\text{-}16$ | *** |
| $\Delta t$ & sailing date | 99 | 280470 | 999697 | 632293 | $< 2.2e\text{-}16$ | *** |
| night sailing & sailing date | 1 | 100481 | 999696 | 531811 | $< 2.2e\text{-}16$ | *** |
| $\Delta t$ & night sailing & sailing date | 99 | 18736 | 999597 | 513075 | $< 2.2e\text{-}16$ | *** |

df: degree of freedom, resid. df.: residual degree of freedom. The interacting variables are connected by &.

sailing by SPVN using dichroic/double-refracting sunstone crystals to the southeast shore of Greenland from Norway. Our new findings are obtained via a computer simulation which is refined compared to its original version [34]. This improved model has subtler features, and therefore, is capable to take into consideration more precisely also the weather that may influence the success of SPVN. One of the main findings of this paper is that SPVN can be very robust against weather conditions, which could be a fundamental factor in verifying the operability and veracity of this navigational method.

Both the earlier [34] and the present simulations of North Atlantic voyages using SPVN share the common feature that the same

1. sunstone crystals (cordierite, tourmaline, calcite),

2. sailing dates (spring equinox, summer solstice) and

3. north error datasets
   were used, furthermore

4. the small quasi-random drift of the ship with lowered sails (when the voyage was stopped at night) due to water currents and wind was neglected.
   In order to make more reliable the model of Viking voyages, the simulations of Száz and Horváth [34] were improved in our present work as follows:

5. The discrete values of the navigation periodicity $\Delta t$ (ranging from 1 to 6 hours with an increment of 1 hour) were changed to continuous values chosen randomly between 0.5 and 6 hours from a uniform distribution.
   The following new variables were introduced:

6. night sailing,

7. cloudiness dominance $m_{\text{dominance}}$ and

8. cloudiness changeability $\sigma_{\text{changeability}}$

9. The sailing speed $w$ of the ship changed around an average value $w_{\text{ave}} = 11$ km/h with a Gaussian normal distribution ($w_{\text{max}} = 21$ km/h, $\Delta w_{\text{ave}} = \pm 2$ km/h), instead of using the earlier constant value (11 km/h).

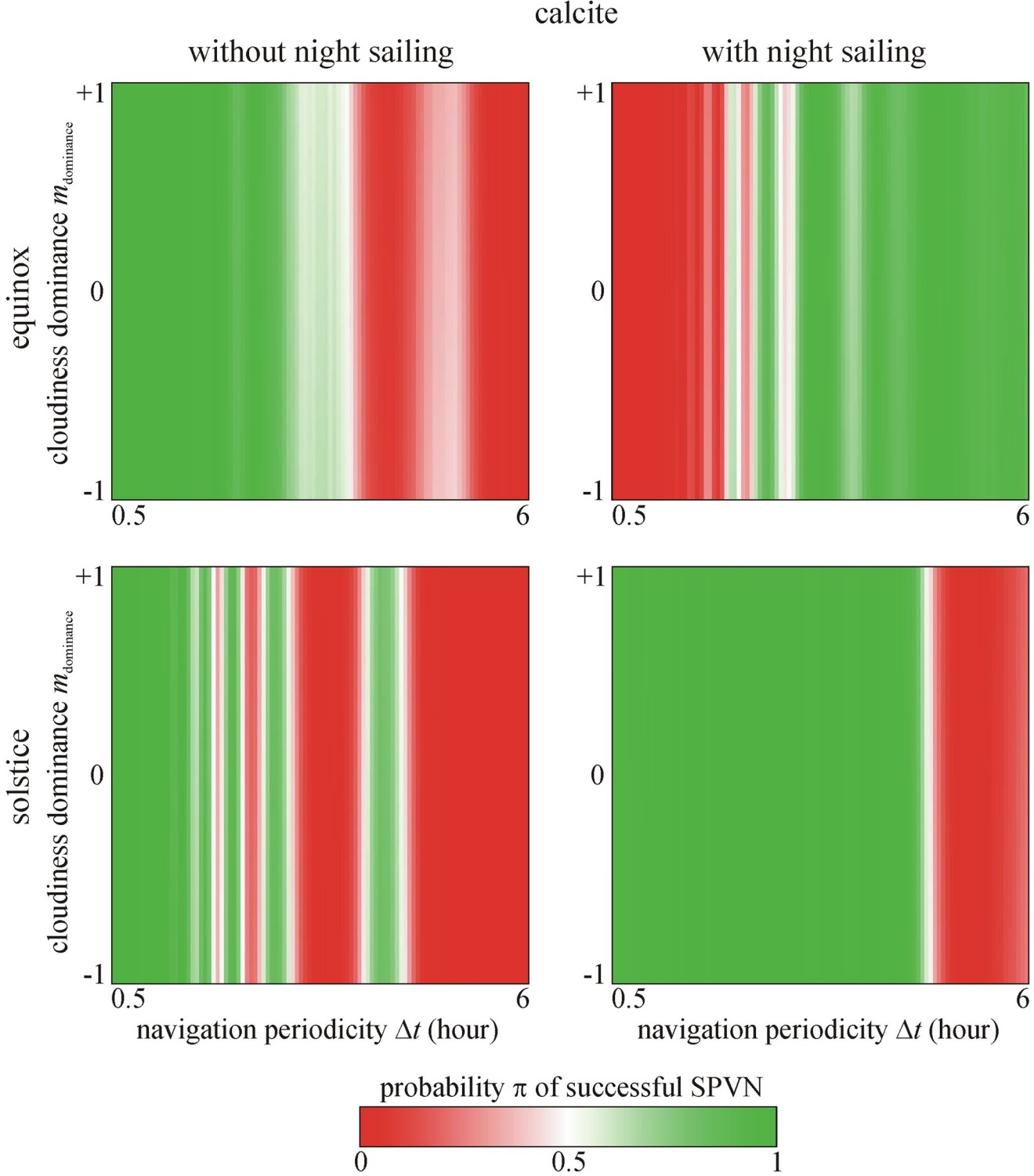

**Fig 4. Probability $\pi$ of successful sky-polarimetric Viking navigation (SPVN) as functions of the navigation periodicity $\Delta t$ and cloudiness dominance $m_{\text{dominance}}$ for calcite sunstone predicted by the 2nd logistic model.** It is clearly seen that $\pi$ is independent of $m_{\text{dominance}}$.

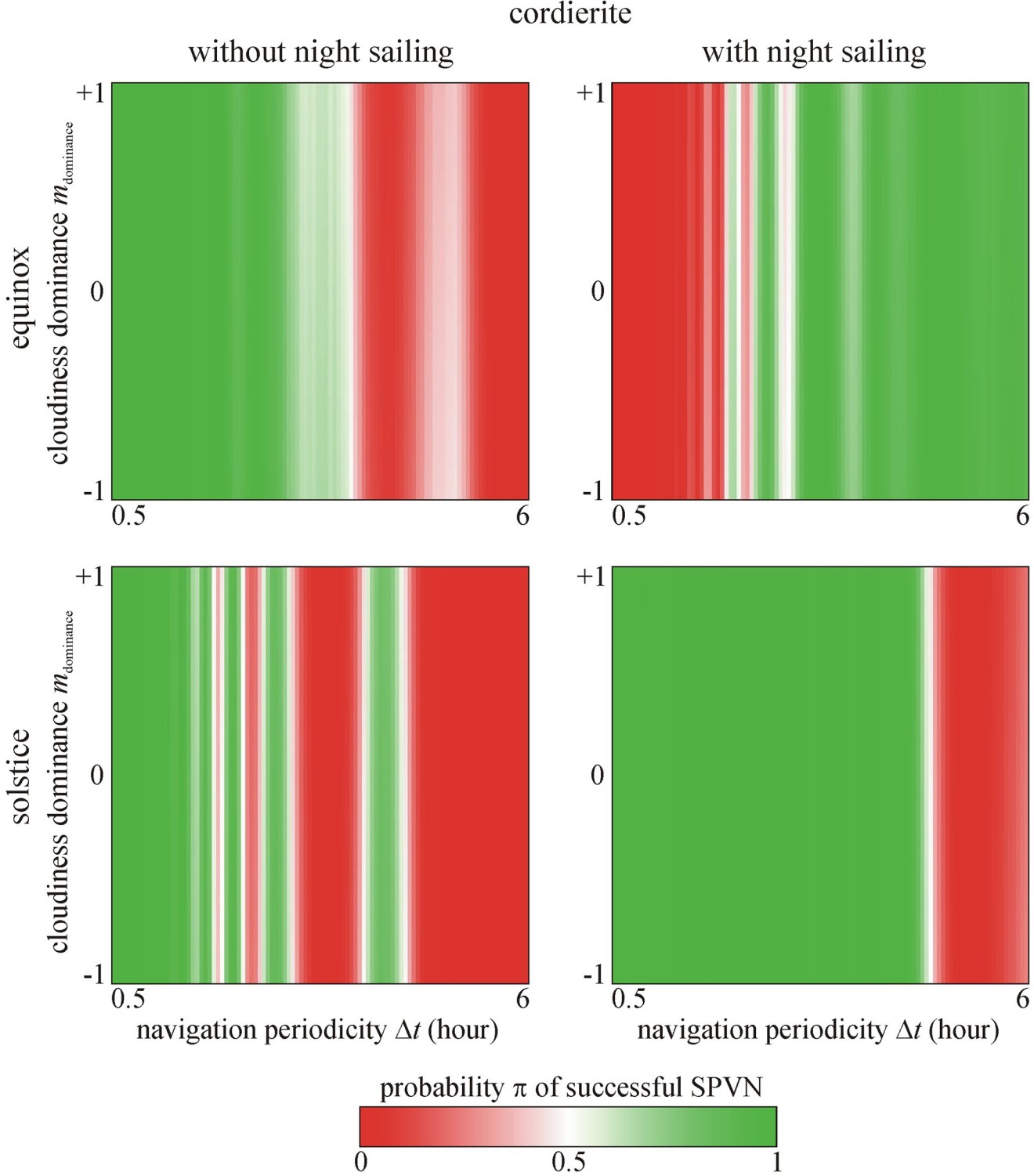

**Fig 5. Probability π of successful SPVN as functions of the navigation periodicity Δ*t* and cloudiness dominance $m_{\text{dominance}}$ for cordierite sunstone predicted by the 2nd logistic model.** It is clearly seen that π is independent of $m_{\text{dominance}}$.

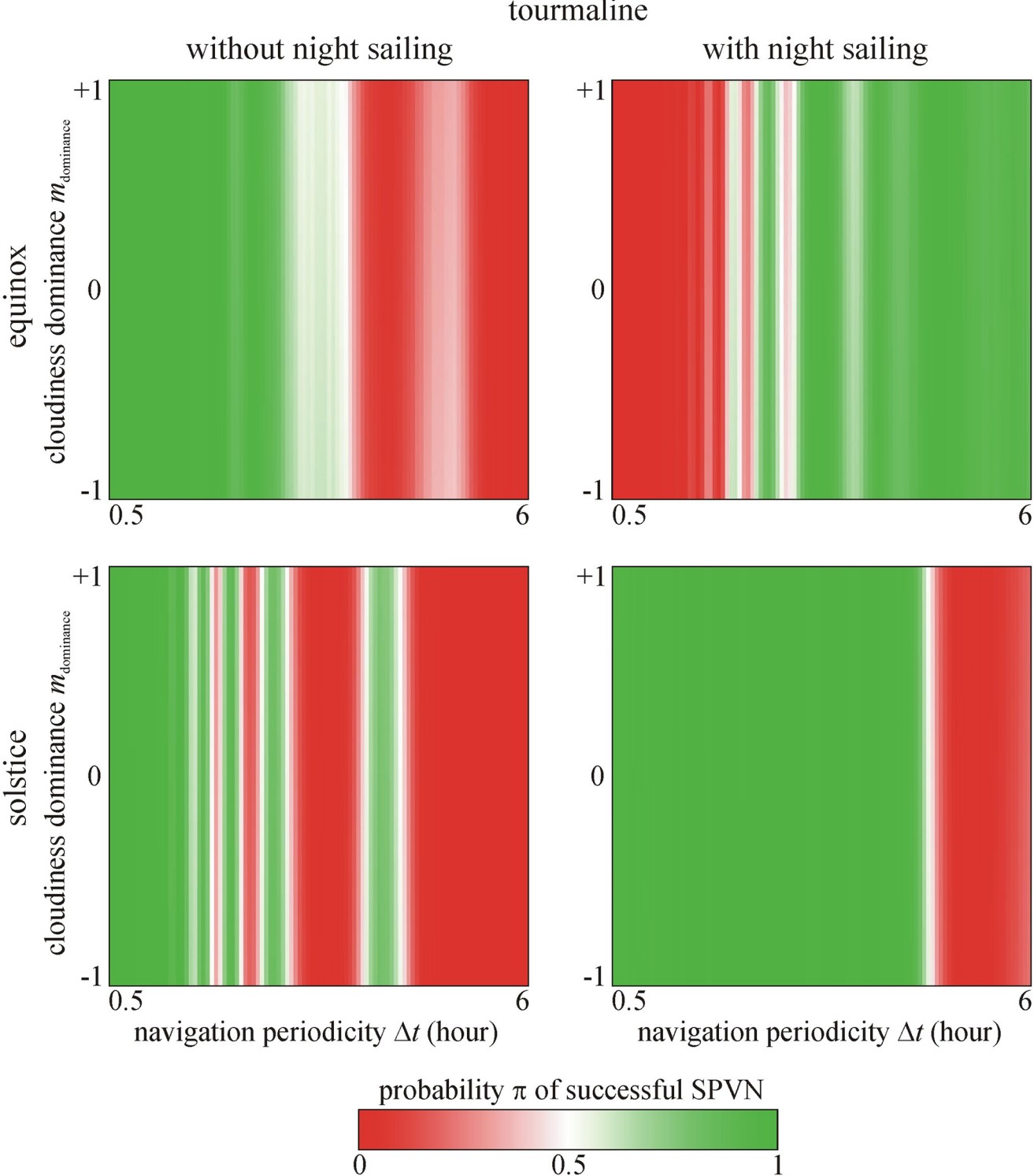

**Fig 6. Probability $\pi$ of successful SPVN as functions of the navigation periodicity $\Delta t$ and cloudiness dominance $m_{\text{dominance}}$ for tourmaline sunstone predicted by the 2nd logistic model.** It is clearly seen that $\pi$ is independent of $m_{\text{dominance}}$.

10. Instead of using the earlier constant visibility distance $d$ of the south coastline of Greenland, $d$ of Greenland's southeast coastline changed randomly between 0 and the maximum value of 128.23 km, which is the theoretical limit of visibility considering the height of the ship. $d = 0$ km and $d = 128.23$ km mean that the weather is extremely cloudy/foggy ($\rho = 8$ okta) and absolutely clear ($\rho = 0$ okta), respectively.

11. The start time of diurnal sailing changed randomly between 0 and 1 hour after sunrise, instead of the earlier constant zero value (start at sunrise).

12. The coastline of Greenland as the direct goal of voyages became limited to a southeast section, opposite to the earlier south borderline (Fig 1).

The aim of this work was to determine the sensitivity/robustness of the success of sky-polarimetric Viking navigation (SPVN) to variables sunstone type, sailing date, navigation periodicity $\Delta t$, night sailing, cloudiness dominance $m_{\mathrm{dominance}}$ and cloudiness changeability $\sigma_{\mathrm{changeability}}$. Száz and Horváth [34] have already shown that SPVN is robust (almost insensitive) against the change of sunstone type, but depends strongly on the sailing date and navigation periodicity. Using a more sophisticated, improved simulation of SPVN, here we corroborated these earlier findings and demonstrated that SPVN is also very sensitive to night sailing, but is robust against the changing weather conditions (dominance and changeability of cloudiness).

If we look at the north error dataset published in [31] and used also in this work, we can see that there are periods within a day, when the sign of this error changes from negative to positive. Negative and positive north error means that the ship will steer more towards south and north, respectively. The success of a voyage is basically depending on whether or not the ship sails longer periods with southern or northern components of the moving direction. Thus, the ratio between these two direction increments will give the net average direction. For example, if a simulated navigator measures the intended sailing direction when the simulation uses a randomly set negative north error just before a period with a positive north error, then the ship will move towards south, and it will only move towards north again after a new measurement of the sailing direction. This situation can create such simulation days, when the ship advances more towards south than north (Fig 7). This is the reason why the navigation periodicity, sailing date and night sailing have a profound effect on the sailing success (Fig 7).

In our simulations, variable $m_{\mathrm{dominance}}$ drives the increment/decrement of cloudiness $\rho$ (okta) in skewing the change of $\rho$-values towards more or less negative or positive changes. In order that this scheme can work without automatically converging to and then getting stuck at zero cloudiness ($\rho = 0$ okta) or total cloudiness ($\rho = 8$ okta), variable $\sigma_{\mathrm{changeability}}$ is needed (with an appropriately greater range of values than that of $m_{\mathrm{dominance}}$) to allow variation to the preset cloudiness dominance. As examples, Fig 8 shows the cloudiness distributions during simulated voyages using cloudiness dominances $m_{\mathrm{dominance}} = -1, 0$ or $+1$ (meaning dominantly weakly, medium or strongly cloudy skies) and cloudiness changeabilities $\sigma_{\mathrm{changeability}} = 1, 2$ or 4 (meaning weakly/slowly, medium or strongly/rapidly changing cloudiness).

It is striking that the success rate $s$ of simulated sailing routes are practically independent of the dominance $m_{\mathrm{dominance}}$ and changeability $\sigma_{\mathrm{changeability}}$ of cloudiness (Figs 2B and 4–6). This can partly be explained with the cumulated navigation (north) error due to cloudiness having only a low scatter near the end point of the sailing route so that it practically does not change the success rate $s$. Note that $s$ depends mainly on the (a)symmetry of the particular north error cumulated in the morning and the afternoon of every day. Using the same data series as we in this work, Száz et al. [31] have already studied the dependence of the north error on the cloudiness $\rho$, taking into consideration the solar elevation and the sailing date (spring equinox or

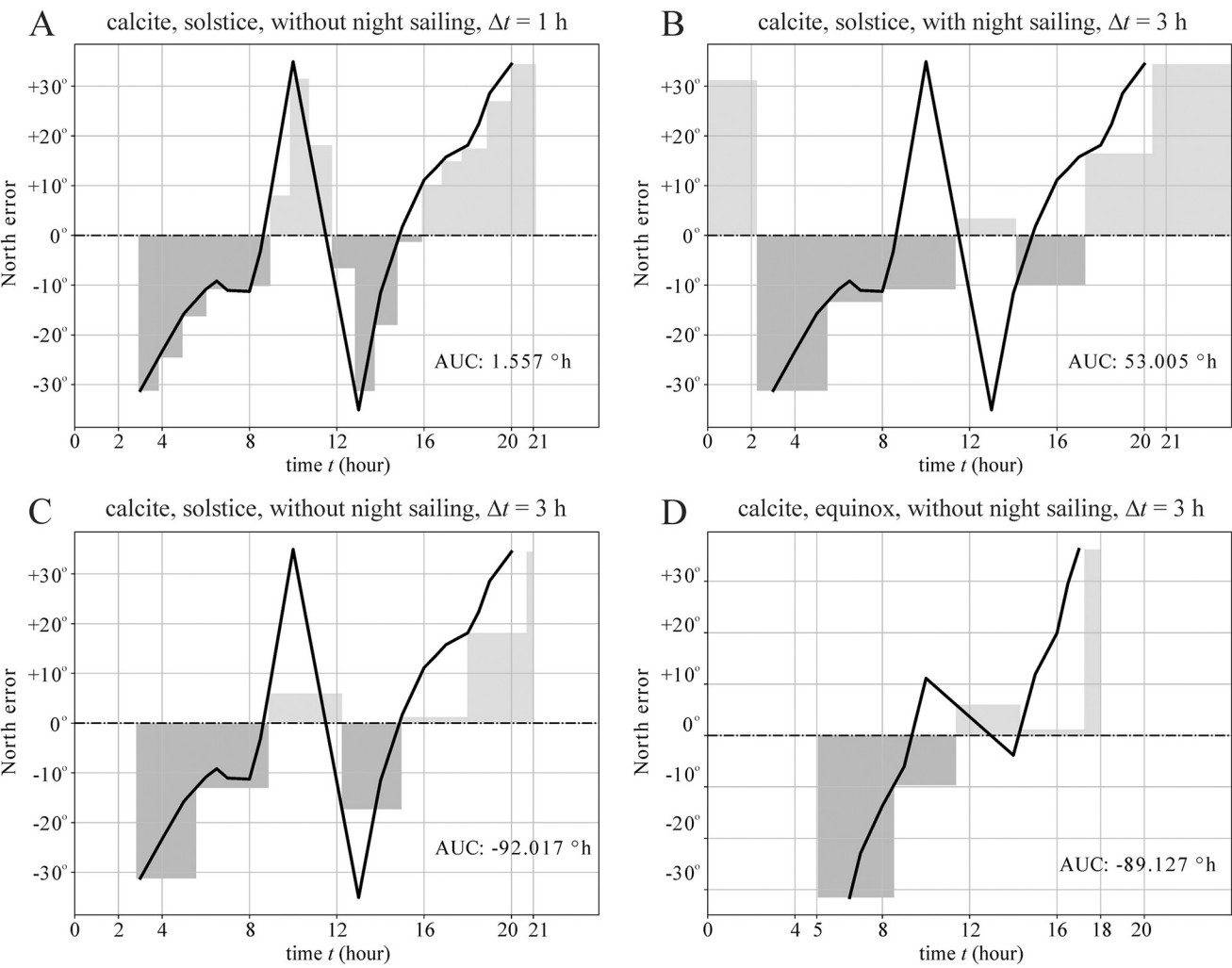

**Fig 7. North error accumulated on a day of sailing versus time in a day at solstice (A, B, C) and equinox (D) using calcite sunstone.** The black curve on each figure represents the north error averaged for cloudiness as a function of time. Each rectangular column represents a sailing period where the area of the rectangle is the accumulated north error for a measurement cycle. The area under the curve AUC is the averaged north error a simulated Viking route suffers on that day (= sum of the area of rectangles).

summer solstice). They also found (see Fig 4 of [31]) that both the net (cumulated) morning and the net afternoon north errors are practically independent of ρ, but depend on the sailing date (spring equinox or summer solstice), the solar elevation and the time of day (morning or afternoon). Hence, since the north error does not vary significantly with changing ρ, ρ has no great effect on the morning-afternoon asymmetry, and therefore it influences only slightly the success rate $s$ of SPVN.

The night sailing results in a big difference compared to the situation without night sailing, since at night the ship advances towards the last direction measured immediately before sunset. This could create situations when the ship steers towards a bad direction (deviating from the intended sailing direction), however, it can also create situations when the direction errors made during the day are actually compensated. Fig 1B, 1D and 1F show that the simulated voyages with night sailing headed more northward compared to those without night sailing (Fig 1A, 1C and 1E). The reason for this is that in afternoon the direction error heads tendentiously

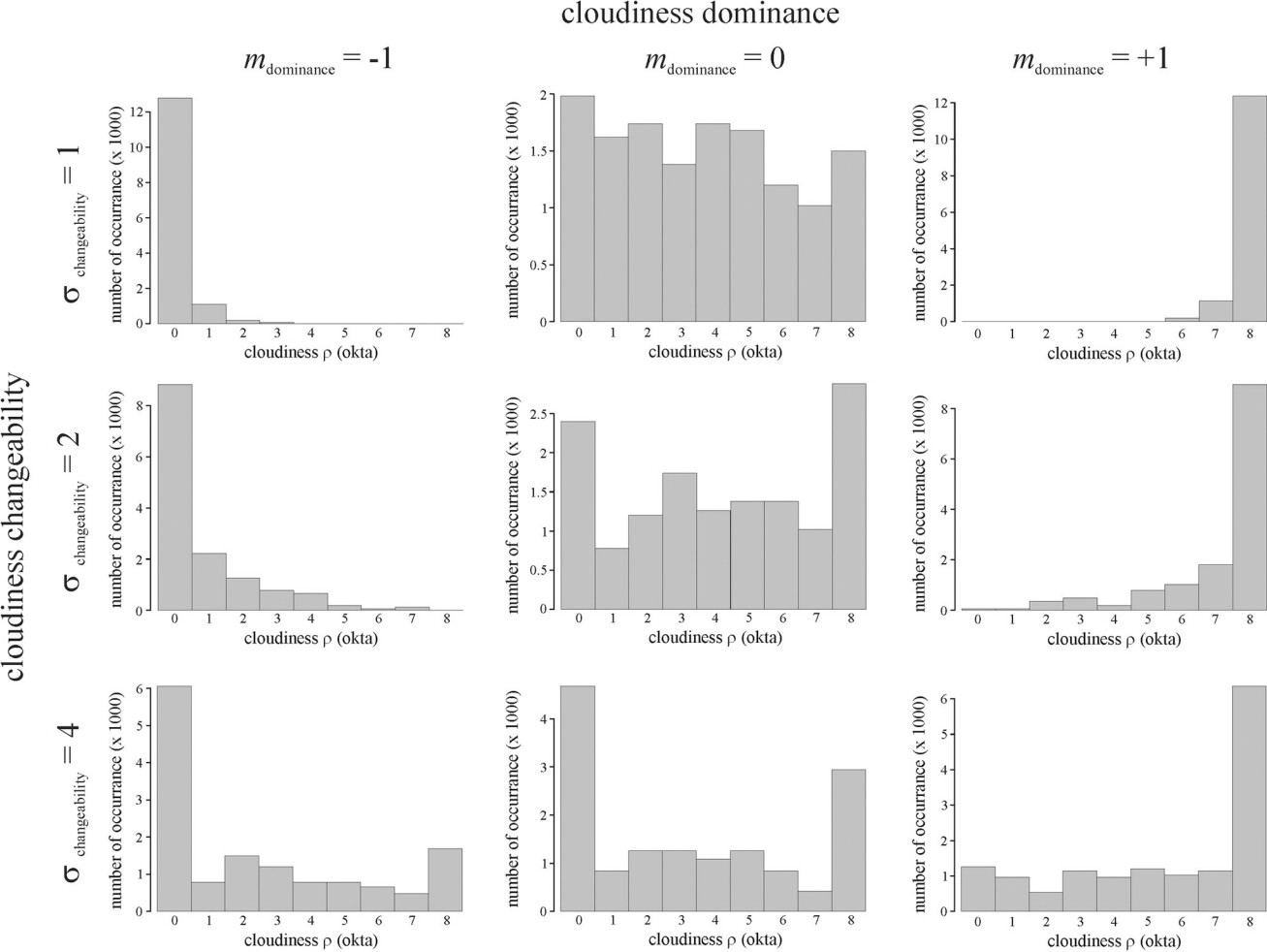

**Fig 8. Cloudiness distributions of Viking sailing routes.** Cloudiness distributions during simulated voyages using calcite sunstones at summer solstice with navigation periodicity $\Delta t$ = 1 hour for cloudiness dominances $m_{dominance}$ = -1, 0 or 1 (meaning dominantly weakly, medium or strongly cloudy skies) and for cloudiness changeabilities $\sigma_{changeability}$ = 1, 2 or 4 (meaning weakly/slowly, medium or strongly/rapidly changing cloudiness).

northward [31], and with night sailing the simulated ship continued its way to this northward direction all night.

The graphs in Figs 2 and 3A–3C show smaller- or higher-amplitude oscillation of the success rate $s$ versus the navigation periodicity $\Delta t$ for certain periods with lower and higher local extrema. The main reason for this phenomenon is the alternating sign of the strongly oscillating north error accumulated on a day (Fig 7): At higher navigation periodicities than 1 hour, the absolute accumulated north error can greatly vary, depending on the actual point of time of navigation. Thus, for a certain periodicity $\Delta t$ the compensation for the accumulated north error can be good, while for a slightly different $\Delta t$-value it can break down quickly by accumulating north error skewed into one of northern or southern directions.

In the future, it could be studied how a progressive restriction of the coastline section defined as successful destination influences the success rate $s$ of Viking navigation. It is expected that $s$ gradually decreases as the desired destination becomes more and more localized. Thus, the chance of a precise localization of the destination is small in all probability.

## Conclusion

We found that among the investigated meteorological parameters and navigation/sailing variables, the success of SPVN is robust against the sunstone type, as well as the cloudiness dominance and changeability. Contrary to this, SPVN is sensitive to the sailing date, navigation periodicity and night sailing, which can optimally be chosen/selected by the sailors/navigators. Remarkably, the accuracy of this navigation method is practically not affected by the dominance and changeability of cloudiness, though one could expect that weather plays the largest role in the sailing success.

## Acknowledgments

We thank two anonymous referees for their constructive reviews.

## Author Contributions

**Conceptualization:** Péter Takács, Dénes Száz, Ádám Pereszlényi, Gábor Horváth.

**Data curation:** Gábor Horváth.

**Formal analysis:** Péter Takács, Dénes Száz, Ádám Pereszlényi, Gábor Horváth.

**Funding acquisition:** Gábor Horváth.

**Investigation:** Péter Takács, Dénes Száz, Ádám Pereszlényi, Gábor Horváth.

**Methodology:** Péter Takács, Dénes Száz, Ádám Pereszlényi, Gábor Horváth.

**Project administration:** Gábor Horváth.

**Resources:** Gábor Horváth.

**Software:** Péter Takács, Dénes Száz.

**Supervision:** Gábor Horváth.

**Validation:** Péter Takács, Dénes Száz, Ádám Pereszlényi, Gábor Horváth.

**Visualization:** Péter Takács, Dénes Száz, Gábor Horváth.

**Writing – original draft:** Péter Takács, Dénes Száz, Ádám Pereszlényi, Gábor Horváth.

**Writing – review & editing:** Gábor Horváth.

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
