## [Decision Letter · Decision Letter 0]

29 Oct 2021

PONE-D-21-15021Sensitivity and robustness of sky-polarimetric Viking navigation Sailing success is most sensitive to night sailing, navigation periodicity and sailing date, but robust against weather conditionsPLOS ONE

Dear Dr. Horvath,

Thank you for submitting your manuscript to PLOS ONE. After careful consideration, we feel that it has merit but does not fully meet PLOS ONE’s publication criteria as it currently stands. Therefore, we invite you to submit a revised version of the manuscript that addresses the points raised during the review process.

Dear authors, please take carefully into consideration the suggested minor revisions in editing the new version of the manuscript.  Please submit your revised manuscript by Dec 13 2021 11:59PM. If you will need more time than this to complete your revisions, please reply to this message or contact the journal office at plosone@plos.org. Please include the following items when submitting your revised manuscript:A rebuttal letter that responds to each point raised by the academic editor and reviewer(s). You should upload this letter as a separate file labeled 'Response to Reviewers'.A marked-up copy of your manuscript that highlights changes made to the original version. You should upload this as a separate file labeled 'Revised Manuscript with Track Changes'.An unmarked version of your revised paper without tracked changes. You should upload this as a separate file labeled 'Manuscript'.

We look forward to receiving your revised manuscript.

Kind regards,

Simone Lolli

Academic Editor

PLOS ONE

Additional Editor Comments (if provided):

Dear authors, please take carefully into consideration the suggested minor revisions in editing the new version of the manuscript before publication.

Journal Requirements:

3. Please include in your Methods section details of how the model was implemented. If the code has been or can be publicly shared, please also include information on where the code can be found, both in the Methods section and the Data availability statement.

4. We note that Figure 1 in your submission contain map images which may be copyrighted. All PLOS content is published under the Creative Commons Attribution License (CC BY 4.0), which means that the manuscript, images, and Supporting Information files will be freely available online, and any third party is permitted to access, download, copy, distribute, and use these materials in any way, even commercially, with proper attribution. For these reasons, we cannot publish previously copyrighted maps or satellite images created using proprietary data, such as Google software (Google Maps, Street View, and Earth). For more information, see our copyright guidelines: http://journals.plos.org/plosone/s/licenses-and-copyright.

Reviewers' comments:

Reviewer's Responses to Questions

**Comments to the Author**

1. Is the manuscript technically sound, and do the data support the conclusions?

Reviewer #1: Yes

Reviewer #2: Yes

2. Has the statistical analysis been performed appropriately and rigorously? 

Reviewer #1: Yes

Reviewer #2: Yes

3. Have the authors made all data underlying the findings in their manuscript fully available?

Reviewer #1: Yes

Reviewer #2: Yes

4. Is the manuscript presented in an intelligible fashion and written in standard English?

Reviewer #1: Yes

Reviewer #2: Yes

5. Review Comments to the Author

Reviewer #1: Title: “Sensitivity and robustness of sky-polarimetric Viking navigation Sailing success is most sensitive to night sailing, navigation periodicity and sailing date, but robust against weather conditions”

Authors: Péter Takács, Dénes Száz, Ádám Pereszlényi and Gábor Horváth

submitted to PLoS One.

General comments

In the last decades, Horvath and colleagues have published a remarkable series of articles dealing with the atmospheric optical and psychophysical aspects of the so-called sky-polarimetric Viking navigation. In their last Viking paper [Száz D, Horváth G. (2018) Success of sky-polarimetric Viking navigation: Revealing the chance Viking sailors could reach Greenland from Norway. Royal Society Open Science 5: 172187 (doi: 10.1098/rsos.172187)], they demonstrated the high success rate of this navigational method which, however, depends on several sailing, meteorological and nautical, navigational parameters. Thus, until now it was unknown how variations in these control parameters might influence the navigation error. In this present manuscript before me, the authors studied the sensitivity of the success of sky-polarimetric Viking navigation in relation to sunstone type, sailing date, navigation periodicity, night sailing, dominance of strongly, medium or weakly cloudy skies, and change in cloud cover. They showed that the sailing success was robust against the sunstone type, dominance of strongly, medium or weakly cloudy skies, and cloud cover changeability, but sensitive to night sailing, navigation periodicity and sailing date. With these results they significantly contributed to the field of the ancient voyages generally and the enigmatic Viking navigation in particular.

The scientific problem addressed is intriguing, the literature review is thorough, the used methods and statistical analyses are relevant and correct, the results are new and important, and their interpretation is convincing. I recommend that the paper be published in PLoS One, subject to some minor revision along my specific comments.

Specific comments

There are some very minor issues with grammar and style, but to improve the readability of the ms I suggest the authors go over their ms once again carefully, preferably with someone with an excellent command of English unless the journal assists in this matter. I am giving some example of some minor corrections below:

Line 27: “…did not have magnetic compasses” or “ did not have a magnetic compass…”

L 28: “…under a sunny sky…”

L37: “…which had a strong and a weak influence…” or “which had strong and weak influences..”

L40: “…while it is robust…”

L42: “…and changing cloudiness…” or “..changing cloud cover..”

L51: “…without a magnetic compass..!”’

L52:”…under sunny weather conditions…”

L56: delete “their”

L82. “…have a strong and weak…”

Questions

1. Why were the voyages simulated only at spring equinox and summer solstice? What about the other days of the year?

2. The authors wrote: The simulated Viking ship moved in a constant direction along a straight line with a constant (but varying) speed...

How can the speed be constant, if it is varying? Please clarify this.

3. The authors wrote: The angle of deviation from the geographical north, called north error, was determined as described in [34].

Please summarize briefly this determination in order that the readers do not need to consult with article [34].

4. Do the authors plan to continue their study with such a computer simulation in which they compare the success of Viking navigation when the navigator takes into consideration not only the sky polarization, but also or only the Sun, if it is visible? This would be important, because during a several-week Viking voyage the Sun was surely visible for many occasions, thus in sunshine the navigator did not need to use the polarization pattern of the sky. This possibility should briefly be addressed in the Discussion section.

5. Would it have made much difference if the departure had not been from Bergen, but further north or south in Norway? Was that because of prevailing ocean currents that were taken into consideration?

Reviewer #2: The paper represents an important sequel to previous studies done by the authors in the topic of sky-polarimetric Viking navigation. It provides further elaboration on the potential success rates of sailing to the shores of Greenland from Norway by using a polarization-based navigation technique with dichroic crystals.

More specifically, here the authors present their findings obtained via a simulation model that has been refined compared to its original version used in their former study. This new model has more subtle features, and as such, it is capable of providing a more accurate representation of relevant factors, including weather, that may influence the success of polarization-based sailing.

One of the main finding of the paper is that navigation by polarization can be very robust against weather conditions, which could be a fundamental factor in verifying the operability and veracity of sky-polarimetric Viking navigation.

In my view the findings of the paper are clear and presented well, there are only minor points that would require a slight revision. These are the following:

1. Pg. 3., lines 142-149: "Each succeeding ρ-value was calculated by adding a discrete change (increment) in okta to the previous ρ-value according to a second Gaussian distribution with median m_dominance and deviation σ changeability, […]"

The authors here describe how the cloudiness is set and changed along a given sailing route. One of the introduced parameters is cloudiness dominance 'm_dominance'. If I understand well, then 'm_dominance' solely drives the increment or decrement of the ρ-value (okta) in skewing the change of ρ-values towards more or less negative or positive changes. However, in order that this scheme work without automatically converging to and then getting stuck at zero cloudiness or total cloudiness, cloudiness changeability 'σ_changeability' is needed (with a sufficiently greater range of values than those of 'm_dominance') to allow variation to the preset cloudiness dominance.

In my opinion this should be explained somewhat in more detail in the paper so that it be completely clear what are the consequences of using these particular parameters and their available ranges.

In fact, it might be even better to check and see that along a given route what is the average and deviation of ρ-values when using a given 'm_dominance' and 'σ_changeability', and among all simulated routes what is their distribution.

2. Pg. 4. lines 153-154: "The simulated Viking ship moved in a constant direction along a straight line with a constant (but varying) speed until […]"

This sentence would require some rewording, as "constant (but varying) speed" is a contradictory statement. I suppose what is meant here is that sailing speed was constant during a given time period, but across time segments it could be varied.

3. Fig.2 and Fig.5, 6.: It is striking that the success rates of simulated sailing routes are practically independent of cloudiness dominance 'm_dominance', as it is also described by the authors in the Results and Discussion parts. However, this would warrant some deeper explanation in my view. I would assume that this has more or less something to do with the cumulated navigation error due to cloudiness having such a low scatter in the position of the sailing route end point so that it practically does not change the success rate in the definition of the paper's model. This would be worth checking against, and it might be also interesting to see a scatter plot of sailing routes (or only their end points) at a given 'm_dominance' value (or in a restricted range).

Here I would also propose an aspect that may be out of scope of this paper but perhaps relevant to some future work: the evaluation of simulation results could include a progressive restriction of the coast line section accepted as successful destination. It could be studied how far this can be restricted (i.e. the desired destination to be more and more localized) and how success rates degrade depending on relevant factors.

4. Fig. 3B,D: There is a clear inversion of the success rate trend against navigation periodicity delta_t between spring equinox and summer solstice, when allowing night sailing. Is there anything that can be said how this would look like at other dates and if there were any periods of the year, when a proper navigation periodicity could be selected for such a sailing mode?

5. Fig. 2. and Fig. 3C,D: the graphs show high amplitude oscillation of success rates against navigation periodicity delta_t for certain periods, with quite low and high local extrema. Can the authors explain what are the reasons of this? I suppose the explanation may partly be given by North error accumulations as displayed on Fig. 7, that is, with higher navigation periodicities than 1 hour, the absolute accumulated error can greatly vary, depending in which discrete points the actual navigation happens, so that for a certain periodicity the compensation of North accumulation error can be good, while for a slightly different periodicity it can break down quickly by accumulating error skewed into one of the directions.

6. PLOS authors have the option to publish the peer review history of their article (what does this mean?). If published, this will include your full peer review and any attached files.

Reviewer #1: **Yes: **Victor Benno MEYER-ROCHOW

Reviewer #2: No

---

## [Author Response · Author response to Decision Letter 0]

16 Nov 2021

Our detailed response to the comments of the two reviewers is uploaded as a separate file (+VikingNavigation_PLoS-One_response.docx).

---

## [Decision Letter · Decision Letter 1]

5 Jan 2022

Sensitivity and robustness of sky-polarimetric Viking navigation Sailing success is most sensitive to night sailing, navigation periodicity and sailing date, but robust against weather conditions

PONE-D-21-15021R1

Dear Dr. Horvath,

We’re pleased to inform you that your manuscript has been judged scientifically suitable for publication and will be formally accepted for publication once it meets all outstanding technical requirements.

Kind regards,

Simone Lolli

Academic Editor

PLOS ONE

Reviewers' comments:

Reviewer's Responses to Questions

**Comments to the Author**

1. If the authors have adequately addressed your comments raised in a previous round of review and you feel that this manuscript is now acceptable for publication, you may indicate that here to bypass the “Comments to the Author” section, enter your conflict of interest statement in the “Confidential to Editor” section, and submit your "Accept" recommendation.

Reviewer #1: All comments have been addressed

Reviewer #2: All comments have been addressed

2. Is the manuscript technically sound, and do the data support the conclusions?

Reviewer #1: Yes

Reviewer #2: Yes

3. Has the statistical analysis been performed appropriately and rigorously? 

Reviewer #1: Yes

Reviewer #2: Yes

4. Have the authors made all data underlying the findings in their manuscript fully available?

Reviewer #1: Yes

Reviewer #2: Yes

5. Is the manuscript presented in an intelligible fashion and written in standard English?

Reviewer #1: Yes

Reviewer #2: Yes

6. Review Comments to the Author

Reviewer #1: Congratulations - this has been a valuable contribution: an interesting paper, well planned and executed..

Reviewer #2: The authors have clearly addressed all outstanding questions and minor issues in a satisfying way, the paper in its now improved form is most definitely suitable for publication.

7. PLOS authors have the option to publish the peer review history of their article (what does this mean?). If published, this will include your full peer review and any attached files.

Reviewer #1: **Yes: **Victor Benno MEYER-ROCHOW

Reviewer #2: No

---

## [Editor Report · Acceptance letter]

10 Jan 2022

PONE-D-21-15021R1 

Sensitivity and robustness of sky-polarimetric Viking navigation: Sailing success is most sensitive to night sailing, navigation periodicity and sailing date, but robust against weather conditions 

Dear Dr. Horvath:

I'm pleased to inform you that your manuscript has been deemed suitable for publication in PLOS ONE. Congratulations! Your manuscript is now with our production department. 

Kind regards, 

on behalf of

Dr. Simone Lolli 

Academic Editor

PLOS ONE